# Uncertainty Propagation for the Structures with Fuzzy Variables and Uncertain-but-Bounded Variables

**DOI:** 10.3390/ma16093367

**Published:** 2023-04-25

**Authors:** Yanjun Xia, Linfei Ding, Pan Liu, Zhangchun Tang

**Affiliations:** 1School of Mechatronics Engineering, University of Electronic Science and Technology of China, Chengdu 611731, China; 2Nanjing Research Institute of Simulation Technology, Nanjing 210016, China; 3Tianfu Innovation Energy Establishment, Chengdu 610000, China

**Keywords:** uncertainty propagation, uncertain-but-bounded variables, fuzzy variables, non-probabilistic reliability index, multi-ellipsoid convex set

## Abstract

Various uncertain factors exist in the practical systems. Random variables, uncertain-but-bounded variables and fuzzy variables are commonly employed to measure these uncertain factors. Random variables are usually employed to define uncertain factors with sufficient samples to accurately estimate probability density functions (PDFs). Uncertain-but-bounded variables are usually employed to define uncertain factors with limited samples that cannot accurately estimate PDFs but can precisely decide variation ranges of uncertain factors. Fuzzy variables can commonly be employed to define uncertain factors with epistemic uncertainty relevant to human knowledge and expert experience. This paper focuses on the practical systems subjected to epistemic uncertainty measured by fuzzy variables and uncertainty with limited samples measured by uncertain-but-bounded variables. The uncertainty propagation of the systems with fuzzy variables described by a membership function and uncertain-but-bounded variables defined by a multi-ellipsoid convex set is investigated. The combination of the membership levels method for fuzzy variables and the non-probabilistic reliability index for uncertain-but-bounded variables is employed to solve the uncertainty propagation. Uncertainty propagation is sued to calculate the membership function of the non-probabilistic reliability index, which is defined by a nested optimization problem at each membership level when all fuzzy variables degenerate into intervals. Finally, three methods are employed to seek the membership function of the non-probabilistic reliability index. Various examples are utilized to demonstrate the applicability of the model and the efficiency of the proposed method.

## 1. Introduction

Based on classical probability theory, traditional probabilistic reliability analysis has been more and more perfect. The main purpose of probabilistic reliability analysis is to assess reliability or failure probability. Many practical methods, such as Monte Carlo simulation, the importance sampling method [1], the response surface method [2], the first-order reliability method (FORM) [3], the second-order reliability method (SORM) [4], the subset simulation [5], the directional method [6], the line sampling method [7] and the asymptotic method for SORM [8], have been proposed to achieve this aim and apply it to practical engineering problems.

However, the traditional probabilistic reliability model requires precise probability density functions of the random variables, which are difficult to obtain in many practical applications because the samples available in practical engineering problems are limited. Although the principle of maximum entropy has been employed as an efficient technique to model the concerned uncertainty with a probabilistic distribution [9], it has been pointed out that classical probability reliability may be extremely sensitive to the statistical distribution of the data and even small errors in the inputs may yield misleading results in some cases [10,11]. This implies that the traditional probabilistic reliability model may be unable to deal with some problems with incomplete information (or limited samples of the inputs). Fortunately, many novel strategies such as the non-probabilistic model [10,11,12,13,14,15,16,17,18,19,20], fuzzy variables and the fuzzy uncertainty propagation model [21,22,23,24] as well as the fuzzy randomness model [25,26] have been provided to deal with such cases.

Compared with the precise probability density functions of the uncertain variables, the bounds of the variables for many engineering problems, however, may be difficult to obtain from the information available. Recently, non-probabilistic reliability models such as the convex set, the interval set and the fuzzy set have been presented as attractive supplements to the traditional probabilistic reliability model [16]. Since the convex set models, including ellipsoid and hyper-box models, were first proposed by Ben-Haim and Elishakoff to describe the uncertain-but-bounded variables [10,11,12,13,14,15], non-probabilistic techniques have become popular in non-deterministic dynamic finite element analysis [17] and the capabilities of the approaches have been discussed in detail [16]. Recently, the multi-ellipsoid convex set model was also proposed to deal with the case when the uncertain-but-bounded variables can be classified into many uncorrelated groups and each group can be defined by a single ellipsoid convex set [18,19,20]. As revealed in several studies [18,19,20], the multi-ellipsoid convex model can be regarded as the extension of the single ellipsoid and hyper-box model—in other words, the single ellipsoid and hyper-box model are two specific instances of the multi-ellipsoid convex set model. According to the non-probabilistic reliability concept stated by Ben-Hain and Elishakoff [10,11,12,13,14,15], the non-probabilistic reliability index of the multi-ellipsoid convex set can be expressed as the maximum allowable variability of the systems, which can be determined by the infinity norm of the vector consisting of Euclidean norms of the uncorrelated group uncertain-but-bounded vectors [18,19,20]. As for the fuzzy variables described by membership functions, the uncertainty propagation in mechanical systems has been investigated widely [21,22] and the membership levels method [23,24] has been employed as an efficient means to evaluate problems with fuzzy uncertainties. The basic principle of the membership levels method, which is used in this paper, is that at each membership level, each fuzzy variable reduces into an interval with a lower and an upper bound, and then the bounds of the output responses can be obtained by optimization or any other technique [23,24]. In other words, if the fuzzy variables are depicted by membership functions, the membership functions of the responses can be approximated by the membership levels method. This is termed uncertain propagation with fuzzy variables. In addition, recently, the previous models have been also employed to deal with various degrees of uncertainty in practical engineering problems [27,28,29,30,31,32,33,34]. An improved dimensional approach to multidisciplinary interval uncertainty analysis is developed in which the extreme values of each interval variable used to determine the system response boundaries are solved using Chebyshev polynomial approximation and an iterative criterion [27]. Zhou developed a fault-tree-based system reliability method to predict the failure probability of system components by non-probabilistic interval models [28]. The unknown but constrained parameters were used as interval variables and the eigenvalues of the elastic stiffness matrix, geometric stiffness matrix and uncertain parameters were divided into deterministic and perturbative parts using perturbation theory [29]. Heng et al. proposed a novel dynamic model updating procedure to efficiently update the interval and nonstationary correlation coefficient matrix (NPCCM) of the modal parameters and to establish their accurate and reliable uncertainty bounds [30]. Xu et al. proposed a two-layer dimensional analysis procedure for the fuzzy finite element method to determine the minimum and maximum (min/max) points at zero cut for each slice of the response surface [31]. The fuzzy finite element method was also employed to solve the issues in eigenfrequency and deflection analysis [32], nonlinear free vibration analysis [33] and high dimensional model representation [34].

In many engineering applications, the uncertainty of the systems may result from many different sources. In such cases, the information of some uncertain variables may be abundant and the precise probability distribution functions (PDFs) can be accurately estimated by the data available; whereas in other cases with limited samples, which are not sufficient to ensure the accurate PDFs, may be defined by uncertain-but-bounded variables. In addition, some uncertain factors may be relevant to human knowledge and expert experience, which is commonly considered as epistemic uncertainty and fuzzy variables are fit to this situation. Based on the realistic problems, the models dealing with mixed uncertain variables such as the model with random variables and fuzzy uncertainties [35,36,37,38], the combination of random variables and uncertain-but-bounded variables [39,40,41] and the mixture of random variables and intervals [42,43] have been proposed to overcome the difficulty. As the studies mentioned above reveal, a number of attempts have been made for mixed uncertainties analysis. However, most of the existing papers focus on either the mixed model of the random variables and fuzzy inputs or the combination of the random variables and uncertain-but-bounded variables. For engineering applications with limited available samples and epistemic uncertainty, the combination of uncertain-but-bounded variables and fuzzy variables has an advantage in dealing with such a situation. It is necessary to perform an investigation for uncertain propagation for this case. Based on non-probabilistic reliability theory and the membership levels method, the main goal of uncertain propagation is to estimate the membership function of the non-probabilistic reliability index. According to the basic idea of the membership levels method, since each fuzzy variable defined by a membership function degenerates into an interval with a lower and an upper bound at each membership level, the output response of the structure with fuzzy variables and uncertain-but-bounded variables will be bounded within an interval, where the output response here is the so-called non-probabilistic reliability index. When the membership level varies within the bound [0,1], the same procedure can be performed, and then the membership function of the non-probabilistic reliability index can be estimated.

The paper is structured as follows. Section 2 introduces the non-probabilistic convex set model and non-probabilistic reliability index. The membership levels method estimating the fuzzy variables is briefly given in Section 3. The detailed discussion and calculation for uncertainty propagation with fuzzy variables and uncertain-but-bounded variables are presented in Section 4. Five examples are proposed to demonstrate the applicability of the presented mixed model in Section 5. The conclusion is drawn in Section 6.

## 2. Non-Probabilistic Convex Set Model

Compared with the precise probability distribution, the bounds of the input variables can be easily obtained for most engineering applications. The hyper-box model and the multi-ellipsoid convex set model can be employed to describe these uncertain-but-bounded variables. The uncertain-but-bounded variables are independent and reach the limits simultaneously in the hyper-box model while the uncertainties are correlated with each other in the multi-ellipsoid convex set model. However, it can be seen that the former is the specific instance of the latter whereas the latter is the extension of the former. In this section, the single ellipsoid convex set model is briefly discussed, and then it is further extended to the general multi-ellipsoid convex set model. The mathematical definition of the multi-ellipsoid convex model is described as follows, and more detailed information can be found in [10,11,12,13,14,15,16,17,18,19,20].

### 2.1. Single Ellipsoid Convex Set Model

The hyper-box model (or interval convex set model) is only suitable for the case where all the uncertain-but-bounded variables are uncorrelated and vary independently. However, the uncertain-but-bounded variables are correlated with each other in many engineering problems, and, the hyper-box model may not apply in such cases. The ellipsoid convex set model, which can be described by the following expression according to the ellipsoid convex set theory [12], is verified to be a more reasonable definition for such cases.
(1)x∈E={x|(x−xc)TW(x−xc)≤θ2}
where xc=(x1c,⋯,xnc), which is presumed to lie in the reliable domain in this paper, is the nominal value vector of the uncertain-but-bounded variables; W∈ℝn×n is a symmetric positive definite matrix called the characteristic matrix of the ellipsoid convex set model, which describes the orientation and aspect ratio of the principal axes of the ellipsoid model; θ, which defines the size of the ellipsoid model (or the magnitude of the uncertain-but-bounded variables variability) is a positive real number. In practical engineering problems, these parameters can be obtained from the available data, such as tolerance specifications provided by the producers. When only one uncertain-but-bounded variable is involved in Equation (1), the corresponding ellipsoid convex set model can be expressed as [20]
(2)x∈E={x|(x−xc)TW(x−xc)≤θ2}

Then, Equation (2) can be further simplified into the following form
(3)$$x\in E=\left\{x|x^{c}-\theta /\sqrt{W}\leq x\leq x^{c}+\theta /\sqrt{W}\right\}$$

This means that the hyper-box model with only one uncertain-but-bounded variable can be viewed as the specific instance of the single ellipsoid convex set model. In other words, the interval convex set with only one uncertain-but-bounded variable is a one-dimensional single ellipsoid convex set model [20].

### 2.2. Multi-Ellipsoid Convex Set Model

In practical engineering applications, the considered uncertain-but-bounded variables may arise from different sources such as inaccuracies in geometry, variability in material properties, fluctuations in external loads and errors resulting from instrument measurements. Therefore, it is more reasonable to divide all the uncertain-but-bounded variables into several uncorrelated groups according to the uncertainty source and then the multi-ellipsoid convex model can be established based on these uncorrelated groups, where each group is defined by a sub-dimensional ellipsoid convex set model according to the corresponding uncertainty source. For example, the variability in material properties can be described by an ellipsoid convex set model and errors arising from the instrument measurements by another. Suppose that the uncertain-but-bounded variables can be classified into k uncorrelated groups, and the corresponding vector is defined by
(4)xT={x1T,⋯,xkT}
where xi∈ℝni(i=1,⋯,k) denotes the *i*th group of the uncertain-but-bounded variables, and ni is the total number of uncertainties belonging to the *i*th group, which satisfies the following relation ∑i=1kni=n, where n represents the total number of the uncertain-but-bounded variables.

The single ellipsoid convex set model in Equation (1) can be extended to the multi-ellipsoid convex set model. For the multi-ellipsoid convex set model, each group of uncertain-but-bounded variables can be defined by the following form with an individual ellipsoid convex set model [18,19,20,39,40].
(5)xi∈Ei={xi|(xi−xic)TWi(xi−xic)≤θi2}(i=1,⋯,k)
where Wi∈ℝni×ni is the characteristic matrix of the *i*th ellipsoid convex set model and θi is a positive real number. Wi and θi possess an identical meaning to W and θ in Equation (1). When the number of the groups k is equal to 1, then the multi-ellipsoid convex set model reduces to a single one.

Obviously, if each group consists of only one uncertain-but-bounded variable, similar to Equations (2) and (3), it will reduce to an interval expressed as Equation (6), and then the multi-ellipsoid convex set model will degenerate into a hyper-box model. This implies that the hyper-box convex set model is a specific instance of the multi-ellipsoid convex set model [20].
(6)$$\begin{array}{l} x_{i}\in E_{i}=\left\{x_{i}|{\left(x_{i}-x_{i}^{c}\right)}^{T}W_{i}\left(x_{i}-x_{i}^{c}\right)\leq \theta_{i}^{2}\right\}\\ x_{i}\in E_{i}=\left\{x_{i}|x_{i}^{c}-\theta_{i}/\sqrt{W_{i}}\leq x_{i}\leq x_{i}^{c}+\theta_{i}/\sqrt{W_{i}}\right\} \end{array}$$

A comparison of the three cases of the convex model with three uncertain-but-bounded variables is given in Figure 1, where (a) denotes the hyper-box model with three intervals, (b) represents a single ellipsoid model with three correlated variables and (c) is a multi-ellipsoid model with two sub-dimensional ellipsoids: an interval and a single ellipsoid. Since the single ellipsoid convex set model and hyper-box convex set model are the simplified versions of the multi-ellipsoid model, and thus the non-probabilistic reliability index can first be explained according to the two simple models, and then be further extended to the complex model in the next section.

### 2.3. The Non-Probabilistic Reliability Index

#### 2.3.1. Normalization of Uncertain-But-Bounded Variables

The multi-ellipsoid convex set model in Equation (5) can be transformed into the normalized form expressed in Equation (9) by the standard transformation expressed in Equations (7) and (8) [20].
(7)$$Q_{i}^{T}W_{i}Q_{i}=\Lambda_{i},Q_{i}^{T}Q_{i}=I_{i}$$
(8)$$q_{i}=\left(1/\theta_{i}\right)\Lambda_{i}^{1/2}Q_{i}^{T}\left(x_{i}-x_{i}^{c}\right)\left(i=1,\cdots ,k\right)$$
(9)$$E_{i}=\left\{q_{i}|q_{i}^{T}q_{i}\leq 1\right\}\left(i=1,\cdots ,k\right)$$
where $Q_{i}$ is an orthogonal matrix consisting of the normalized eigenvectors of $W_{i}$, $\Lambda_{i}$ is a diagonal matrix comprising the eigenvalues of $W_{i}$ and $I_{i}$ is a unit matrix, $q_{i}$ is the normalized or standard vector of the *i*th group uncertain-but-bounded vector $x_{i}$. Figure 2 shows the comparison of the three models in the normalized q-space [18].

#### 2.3.2. Non-Probabilistic Reliability Index

##### Single Ellipsoid Convex Set Model

When all the uncertain-but-bounded variables can be described by a single ellipsoid convex set model, the normalized form of the single ellipsoid convex set model can be defined by a hyper-sphere with unit radius according to Equations (7)–(9).
(10)$$E=\left\{q|q^{T}q\leq 1\right\}$$
where $q=(q_{1},q_{2},\cdots ,q_{n})$ and n is the total number of the uncertain-but-bounded variables.

Figure 3 gives a single ellipsoid convex set with two uncertain-but-bounded variables in the standard q-space [18]. The domains surrounded by the dashed-line circles are the expanded convex set. The region encircled by the solid-line circle with unit radius, which is centered at the coordinate origin, represents the convex set formed by all the possible values of the two uncertain-but-bounded variables. According to the basic principle of non-probabilistic reliability [10,11,12,13,14,15,16,17,18,19,20], when the circle enlarges proportionally in two directions, all the possible values of the two uncertain-but-bounded variables will locate in the reliable domain until the circle becomes tangential to the standard limit state curve. The maximum allowable variability, which can be employed to measure the reliability of the systems according to the concept of the non-probabilistic reliability index proposed first by Ben-Hain and Elishakoff [10,11,12,13,14,15], can be determined by the shortest distance from the coordinate origin to the standard limit state curve [18]:(11)$$\begin{array}{l} \eta =\underset{q}{\min}\left\{\sqrt{q^{T}q}\right\}\\ s.t.\;g(q)=0 \end{array}$$
where q is the normalized or standard vector of the uncertain-but-bounded vector x, and min{⋅} is the minimum operation, and g(q)=0 is the normalized failure boundary.

##### Multi-Ellipsoid Convex Set Model

When all the uncertain-but-bounded variables can be classified into k groups and each group consists of only one uncertainty, the multi-ellipsoid convex set model in Equation (9) reduces to a hyper-box model in the q-space, which can be written in the following form.
(12)$$E_{i}=\left\{q_{i}|q_{i}^{2}\leq 1\right\}\left(i=1,\cdots ,k\right)$$
where the normalized uncertain-but-bounded variables $q_{i}=\left(x_{i}-x_{i}^{c}\right)/\Delta x_{i}$.

Figure 4 gives three specific cases of the standard q-space for a structure with two interval variables. Obviously, all the possible values of the two interval variables lie in the domain of the solid-line box, which is centered as the coordinate origin and has a side-length of 2, namely $\left\{-1\leq q_{1}\leq 1,-1\leq q_{2}\leq 1\right\}$. Similar to the procedure of enlarging the convex set boundaries proportionally, the same conclusion can be drawn as follows. For case (a), the maximum variation that the system can tolerate is the value of the vertical coordinate of the critical point A, namely $\underset{q|g(q)=0}{\min}(\max(q_{1},q_{2}))=q_{2}$; For case (b), the allowable maximum variability is $\underset{q|g(q)=0}{\min}(\max(q_{1},q_{2}))=q_{1}$; For case (c), the maximum degree of variability that the structure allows is $\underset{q|g(q)=0}{\min}(\max(q_{1},q_{2}))=q_{1}=q_{2}$. Hence, according to the non-probabilistic reliability theory [10,11,12,13,14,15,16,17,18,19,20], the non-probabilistic reliability index for the case with two intervals can be expressed as [18]
(13)$$\begin{array}{l} \eta =\underset{q}{\min}\left\{\max\left(\left|q_{1}\right|,\left|q_{2}\right|\right)\right\}=\underset{q}{\min}\left\{\max\left(\left|\sqrt{q_{1}^{2}}\right|,\left|\sqrt{q_{2}^{2}}\right|\right)\right\}\\ s.t.\;g(q)=0 \end{array}$$

The result can be further extended to the hyper-box model with k interval variables and the non-probabilistic reliability index is [18]
(14)$$\begin{array}{l} \eta =\underset{q}{\min}\left\{\max\left(\left|\sqrt{q_{1}^{2}}\right|,\left|\sqrt{q_{2}^{2}}\right|,\cdots ,\left|\sqrt{q_{k}^{2}}\right|\right)\right\}\\ \;=\underset{q}{\min}\left\{\max\left(\left|\sqrt{q_{1}^{T}q_{1}}\right|,\left|\sqrt{q_{2}^{T}q_{2}}\right|,\cdots ,\left|\sqrt{q_{k}^{T}q_{k}}\right|\right)\right\}\\ s.t.\;g(q)=0 \end{array}$$

Obviously, the non-probabilistic reliability index, which is employed to measure the safety of the structure, is the infinity norm (L−∞ or maximum norm) of the vector.

Since the multi-ellipsoid convex set model is the extension of the hyper-box interval convex set, and the form for the hyper-box model expressed as Equation (14) can be extended to the non-probabilistic reliability index of the multi-ellipsoid model, which can be defined by
(15)$$\begin{array}{l} \eta =\underset{q}{\min}\left\{\max\left(\sqrt{q_{1}^{T}q_{1}},\cdots ,\sqrt{q_{k}^{T}q_{k}}\right)\right\}\\ s.t.\;g(q)=0 \end{array}$$

## 3. The Membership Levels Method

The membership levels method is usually employed to calculate the fuzzy variables in [24], as shown in Figure 5. Suppose the limit state function of a structure is expressed as M=F(Y), where $Y=(Y_{1},Y_{2},\cdots ,Y_{m})$ is the fuzzy vector defined by the membership functions. At each membership level α, the fuzzy variable $Y_{i}(i=1,\cdots ,m)$ degenerates into a lower and an upper bound (or interval variable) $Y_{i\alpha }\in [{\underline{Y}}_{i\alpha },{\bar{Y}}_{i\alpha }](i=1,\cdots ,m)$. The bounds of the output response M can be calculated by optimization or any other technique. Once all the variables are defined as membership functions, the bounds of the output response M at various a-cuts can be obtained, and then the approximation of the membership functions of the outputs can be obtained. In other words, uncertainty propagation from the fuzzy input $Y_{i}(i=1,\cdots ,m)$ to the output response M can be achieved by the membership levels method.

## 4. Uncertainty Propagation with Uncertain-But-Bounded Variables and Fuzzy Variables

### 4.1. The Membership Function of the Non-Probabilistic Reliability Index

The non-probabilistic reliability index is employed to define the quantified measure of the reliability of the structure with the uncertain-but-bounded variables [18,19,20]. The membership levels method is utilized to calculate the fuzzy variables [24], as shown in Figure 5. However, for many engineering problems with incomplete available information, all the uncertain variables may arise from many different sources such as internal parameters, external loads, etc. Some of these uncertain variables may be uncertain-but-bounded variables, which can be described by the multi-ellipsoid convex set, and others may be fuzzy variables, which can be defined by the membership function. It is necessary to investigate uncertainty propagation within engineering problems with uncertain-but-bounded variables and fuzzy variables. The limit state function of a system with uncertain-but-bounded variables and fuzzy variables is expressed as
(16)$$M_{1}=G(x,Y)$$
where $x^{T}=\left\{x_{1}^{T},\cdots ,x_{k}^{T}\right\}$ represent the k groups of uncertain-but-bounded variables defined by the multi-ellipsoid convex set, and $Y=[Y_{1},\cdots ,Y_{m}]$ are the m fuzzy variables described by the membership functions.

The k groups of uncertain-but-bounded variables $x^{T}=\left\{x_{1}^{T},\cdots ,x_{k}^{T}\right\}$ can be transformed into the normalized ones $q^{T}=\left\{q_{1}^{T},\cdots ,q_{k}^{T}\right\}$ by Equations (7)–(9). At the membership level $\alpha_{i}$, the m fuzzy variables $Y=[Y_{1},\cdots ,Y_{m}]$ can degenerate into m intervals $y_{\alpha_{i}}=\left[y_{1\alpha_{i}},y_{2\alpha_{i}},\cdots ,y_{m\alpha_{i}}\right]$ with lower bounds ${\underline{y}}_{\alpha_{i}}=\left[{\underline{y}}_{1\alpha_{i}},{\underline{y}}_{2\alpha_{i}},\cdots ,{\underline{y}}_{n\alpha_{i}}\right]$ and upper bounds ${\bar{y}}_{\alpha_{i}}=\left[{\bar{y}}_{1\alpha_{i}},{\bar{y}}_{2\alpha_{i}},\cdots ,{\bar{y}}_{n\alpha_{i}}\right]$ by the membership levels method stated in Section 3. Then, the original limit state function $M_{1}=G(x,Y)$ defined in (16) is mapped into the standard one $g(q,y_{\alpha_{i}})$.

In order to state the principal ideas conveniently, the q-space of a problem, which consists of a single ellipsoid convex set with two uncertain-but-bounded variables and a fuzzy variable, is given in Figure 6. In the ellipsoid convex set model introduced in Section 2, the normalized limit state curve g(q)=0 divides the q-space into two parts: the reliable domain and the failure domain, which can be seen in Figure 3 and Figure 4. However, as revealed by Figure 6, the normalized $g(q,y_{\alpha_{i}})=0(y_{\alpha_{i}}\in \left[{\underline{y}}_{\alpha_{i}},{\bar{y}}_{\alpha_{i}}\right])$ consists of a cluster of normalized limit state curves and each single limit state curve corresponds to a possible realization of the intervals $y_{\alpha_{i}}=\left[{\underline{y}}_{\alpha_{i}},{\bar{y}}_{\alpha_{i}}\right]$. In other words, all the possible values of the uncertain-but-bounded variables and the degraded fuzzy variables that satisfy $g(q,y_{\alpha_{i}})=0$ form a banded geometry in the standard q-space. Hence, the q-space is partitioned into three parts: the reliable domain, the critical domain and the failure domain, as shown in Figure 6. Figure 7 gives the case consisting of a hyper-box model with two intervals and a fuzzy variable, the basic idea of which is the same as Figure 6.

Obviously, the shortest distance from the coordinate origin to the normalized limit state curve varies from ${\underline{\eta }}_{\alpha_{i}}$ and ${\bar{\eta }}_{\alpha_{i}}$ as demonstrated in Figure 6 and Figure 7. According to the mathematical definition of the non-probabilistic reliability index described in Section 2, the non-probabilistic reliability index $\eta_{\alpha_{i}}$ for the problems $g(q,y_{\alpha_{i}})=0(y_{\alpha_{i}}\in \left[{\underline{y}}_{\alpha_{i}},{\bar{y}}_{\alpha_{i}}\right])$ varies within the lower and upper bound $\left[{\underline{\eta }}_{\alpha_{i}},{\bar{\eta }}_{\alpha_{i}}\right]$. For the membership level $\alpha_{i}$, uncertainty propagation (or the lower and upper bound of the non-probabilistic reliability index) can be accomplished based on the previous procedure. When the membership level $\alpha_{i}$ varies within the bound $\alpha_{i}\in \left[0,1\right]$, the membership function of the non-probabilistic reliability index can be estimated. The following section will give some approaches to estimating the membership function of the non-probabilistic reliability index.

### 4.2. Estimate the Membership Function of the Non-Probabilistic Reliability Index

Based on the membership levels method, three techniques are introduced to calculate the membership function of the non-probabilistic reliability index. Before all the procedures are performed, the membership level $\alpha_{i}$ is supposed to take a value of $\alpha_{i}=i\times \frac{1}{N}(i=0,1,\cdots ,N)$, where N is the total number of partitions. In order to reduce the computational cost, N takes a value of N=5 in the numerical examples. The following three methods are employed for the normalized limit state curve $g(q,y_{\alpha_{i}})$, where q are normalized uncertain-but-bounded variables and $y_{\alpha_{i}}$ are upper and lower bounds (degenerated fuzzy variables) for y.

#### 4.2.1. Double-Loop Optimization

Based on these properties of the model with fuzzy variables and uncertain-but-bounded variables, the lower ${\underline{\eta }}_{\alpha_{i}}$ and the upper ${\bar{\eta }}_{\alpha_{i}}$ bounds of the non-probabilistic reliability index can be calculated from Equations (17) and (18), as shown in Figure 8:(17)$$\begin{array}{l} {\underline{\eta }}_{\alpha_{i}}=\underset{y_{\alpha_{i}}}{\min}\;\eta (y_{\alpha_{i}})\\ s.t.\;{\underline{y}}_{\alpha_{i}}\leq y_{\alpha_{i}}\leq {\bar{y}}_{\alpha_{i}} \end{array}$$
(18)$$\begin{array}{l} {\bar{\eta }}_{\alpha_{i}}=\underset{y_{\alpha_{i}}}{\max}\;\eta (y_{\alpha_{i}})\\ s.t.\;{\underline{y}}_{\alpha_{i}}\leq y_{\alpha_{i}}\leq {\bar{y}}_{\alpha_{i}} \end{array}$$
where $\alpha_{i}=i\times \frac{1}{N}(i=0,1,\cdots ,N)$ is the value of the membership level, the non-probabilistic reliability index corresponding to $y_{\alpha_{i}}$ is given as the following form
(19)$$\begin{array}{l} \eta (y_{\alpha_{i}})=\underset{q,y_{\alpha_{i}}}{\min}\left\{\max\left(\sqrt{q_{1}^{T}q_{1}},\cdots ,\sqrt{q_{k}^{T}q_{k}}\right)\right\}\\ s.t.g(q,y_{\alpha_{i}})=0 \end{array}$$

The symbols in Equation (19) are identical to the ones in Equation (15).

#### 4.2.2. Single-Loop Optimization

Firstly, the minmax optimization problem expressed as Equation (19) can be transformed into an equivalent minimization problem by introducing a variable δ [19].
(20)$$\begin{array}{l} \eta (y_{\alpha_{i}})=\underset{q,y_{\alpha_{i}},\delta }{\min}\delta \\ s.t.g(q,y_{\alpha_{i}})=0\\ \;q_{i}^{T}q_{i}\leq \delta^{2}(i=1,2,\cdots ,k) \end{array}$$

Then, with the combination of the three sub-optimization problems expressed in Equations (17), (18) and (20), respectively, the lower bound ${\underline{\eta }}_{\alpha_{i}}$ and the upper bound ${\bar{\eta }}_{\alpha_{i}}$ can be equivalently transformed into the single-loop optimization problems expressed as Equations (21) and (22).
(21)$$\begin{array}{l} {\underline{\eta }}_{\alpha_{i}}=\underset{q,\delta ,y_{\alpha_{i}}}{\min}\delta \\ s.t.g(q,y_{\alpha_{i}})=0\\ q_{i}^{T}q_{i}\leq \delta^{2}(i=1,2,\cdots ,k)\\ {\underline{y}}_{\alpha_{i}}\leq y_{\alpha_{i}}\leq {\bar{y}}_{\alpha_{i}} \end{array}$$
(22)$$\begin{array}{l} {\bar{\eta }}_{\alpha_{i}}=\underset{q,\delta ,y_{\alpha_{i}}}{\max}\delta \\ s.t.g(q,y_{\alpha_{i}})=0\\ q_{i}^{T}q_{i}\leq \delta^{2}(i=1,2,\cdots ,k)\\ {\underline{y}}_{\alpha_{i}}\leq y_{\alpha_{i}}\leq {\bar{y}}_{\alpha_{i}} \end{array}$$
where the notations in Equations (21) and (22) are in accordance with the ones in Equations (17) and (18), respectively. The flowchart of computing ${\underline{\eta }}_{\alpha_{i}}$ and ${\bar{\eta }}_{\alpha_{i}}$ is shown in Figure 9.

#### 4.2.3. The Outer Optimization by Random Sampling Method

The only constraints of the outer loop in Equations (17) and (18) are the bounds expressed as ${\underline{y}}_{\alpha_{i}}\leq y_{\alpha_{i}}\leq {\bar{y}}_{\alpha_{i}}$. As revealed in Section 4.1 and in Figure 6 and Figure 7, each possible realization of the intervals $y_{\alpha_{i}}^{j}=\left[{\underline{y}}_{\alpha_{i}},{\bar{y}}_{\alpha_{i}}\right]$ is in accordance with a single normalized limit state curve, where the solid lines (a) and (b) in Figure 6 and Figure 7 represent the normalized limit state curve corresponding to ${\underline{\eta }}_{\alpha_{i}}$ and ${\bar{\eta }}_{\alpha_{i}}$, respectively. In order to estimate the non-probabilistic reliability index ${\underline{\eta }}_{\alpha_{i}}$ and ${\bar{\eta }}_{\alpha_{i}}$, we can first simulate M realizations $y_{\alpha_{i}}^{j}(j=1,2,\cdots ,M)$ within the intervals $\left[{\underline{y}}_{\alpha_{i}},{\bar{y}}_{\alpha_{i}}\right]$ uniformly, and then estimate the non-probabilistic reliability index $\eta (y_{\alpha_{i}}^{j})$ of the normalized limit state curve corresponding to the *j*th realization $y_{\alpha_{i}}^{j}$ by Equation (19) or (20). Finally, the minimum and maximum of the sequences $\eta (y_{\alpha_{i}}^{j})(j=1,2,\cdots ,M)$ can be employed to approximate the lower and upper non-probabilistic reliability index ${\underline{\eta }}_{\alpha_{i}}$ and ${\bar{\eta }}_{\alpha_{i}}$, which can be expressed as Equations (23) and (24). The corresponding flowchart of estimating ${\underline{\eta }}_{\alpha_{i}}$ and ${\bar{\eta }}_{\alpha_{i}}$ is given in Figure 10.
(23)$${\underline{\eta }}_{\alpha_{i}}=\min\left(\eta (y_{\alpha_{i}}^{1}),\eta (y_{\alpha_{i}}^{2}),\cdots ,\eta (y_{\alpha_{i}}^{M})\right)$$
(24)$${\bar{\eta }}_{\alpha_{i}}=\max\left(\eta (y_{\alpha_{i}}^{1}),\eta (y_{\alpha_{i}}^{2}),\cdots ,\eta (y_{\alpha_{i}}^{M})\right)$$

Three methods have been given for estimating the membership function of the non-probabilistic reliability index in Section 4.2.1, Section 4.2.2 and Section 4.2.3. Here, we will discuss the computational cost relevant to the three methods.

For the double-loop optimization method, N=5 membership levels have been employed, i.e., $\alpha_{i}=i\times \frac{1}{N}(i=0,\cdots ,N-1)$. Thus, the total computational cost is $\sum_{i=0}^{N-1}\left[N\left({\underline{\eta }}_{\alpha_{i}}\right)+N\left({\bar{\eta }}_{\alpha_{i}}\right)\right]$, in which $N\left({\underline{\eta }}_{\alpha_{i}}\right)$ is the number of optimization iterations for solving ${\underline{\eta }}_{\alpha_{i}}$ in Equation (17) and $N\left({\bar{\eta }}_{\alpha_{i}}\right)$ is the number of optimization iterations for solving ${\bar{\eta }}_{\alpha_{i}}$ in Equation (18). During every optimization iteration, we need to solve $\eta (y_{\alpha_{i}})$ that is a minimum-maximum nesting optimization in Equation (19), which is a time-consuming process.

For the single-loop optimization method, N=5 membership levels have been employed, i.e., $\alpha_{i}=i\times \frac{1}{N}(i=0,\cdots ,N-1)$. Thus, the total computational cost is $\sum_{i=0}^{N-1}\left[N^{\prime }\left({\underline{\eta }}_{\alpha_{i}}\right)+N^{\prime }\left({\bar{\eta }}_{\alpha_{i}}\right)\right]$, in which $N^{\prime }\left({\underline{\eta }}_{\alpha_{i}}\right)$ is the number of optimization iterations for solving ${\underline{\eta }}_{\alpha_{i}}$ in Equation (21), and $N^{\prime }\left({\bar{\eta }}_{\alpha_{i}}\right)$ is the number of optimization iterations for solving ${\bar{\eta }}_{\alpha_{i}}$ in Equation (22). Obviously, there is no nesting optimization when solving ${\underline{\eta }}_{\alpha_{i}}$ and ${\bar{\eta }}_{\alpha_{i}}$, as shown in Equations (21) and (22), and thus the computational cost relevant to each optimation iteration is low.

For the random sampling method, N=5 membership levels have been employed, i.e., $\alpha_{i}=i\times \frac{1}{N}(i=0,\cdots ,N-1)$. In order to solve ${\underline{\eta }}_{\alpha_{i}}$ and ${\bar{\eta }}_{\alpha_{i}}$, we first generate M realizations $y_{\alpha_{i}}^{j}(j=1,2,\cdots ,M)$ within the intervals $y_{\alpha_{i}}^{j}(j=1,2,\cdots ,M)$ uniformly, and then we can estimate the non-probabilistic reliability index $\eta (y_{\alpha_{i}}^{j})$ of the normalized limit state curve corresponding to the *j*th realization $y_{\alpha_{i}}^{j}(j=1,2,\cdots ,M)$. Thus, the total computational cost is $\sum_{i=0}^{N-1}\sum_{j=1}^{M}N^{\prime \prime }\left[\eta (y_{\alpha_{i}}^{j})\right]$, in which $N^{\prime \prime }\left[\eta (y_{\alpha_{i}}^{j})\right]$ is the computational cost for estimating $\eta (y_{\alpha_{i}})$ given in Equation (19) or (20). In general, the number of realizations (i.e., $y_{\alpha_{i}}^{j}(j=1,2,\cdots ,M)$) is not small, and this paper M takes a value of 1000, i.e., M=1000.

It is obvious that the double-loop optimization method is the most complex procedure, the random sampling method is the second most complex, and the single-loop optimization method is the least complex. The computational cost relevant to the single-loop optimization method can be afforded as the number of uncertainties increases, as shown in Section 5.

## 5. Numerical Examples

It is prohibitive to approximate the membership function of the non-probabilistic reliability index accurately due to the large computational cost. In order to reduce the computational consumption, the lower and upper bounds of the non-probabilistic reliability index corresponding to six membership levels are estimated first, and then the membership function of the non-probabilistic reliability index can be obtained by linking these six values, namely $\alpha_{i}=i\times \frac{1}{N}(i=0,1,\cdots ,5)$. The total number of realizations is 1000 for the random sampling method. The symbol *NOFC* represents the number of function calculations.

### 5.1. A Simple Linear Performance Function

Give a simple performance function $g(x)=x_{1}+x_{2}-x_{3}-2$, where $x_{1}$ and $x_{2}$ are two uncertain-but-bounded variables defined by the following single ellipsoid convex set $x_{1}^{2}+x_{2}^{2}\leq 1$; $x_{3}$ is a fuzzy variable with the following membership function:(25)$$\mu_{x_{3}}(x_{3})=\left\{ \begin{array}{l} \frac{1}{2}x_{3}-\frac{1}{2}\;\;\;\;1\leq x_{3}\leq 3\\ -\frac{1}{2}x_{3}+\frac{5}{2}\;\;3\leq x_{3}\leq 5 \end{array}\right.$$
where $\mu_{x_{3}}(x_{3})$ is the membership level. Table 1 and Table 2 summarize the estimation of the membership function of the non-probabilistic reliability index by single-loop optimization and double-loop optimization, respectively. Figure 11 shows a comparison of the results achieved using the two methods.

This is a simple linear problem, and single-loop optimization and double-loop optimization have obtained an accurate estimator of the membership function of the non-probabilistic reliability index. The results show that the single-loop optimization method and double-loop optimization method can give good results for the linear performance function.

### 5.2. Fourth-Order Polynomial Performance Function

Give a nonlinear performance function $g(x)=\frac{1}{40}x_{1}^{4}+2x_{2}^{2}+x_{3}+3$, where $x_{1}$ is a fuzzy variable described by the following membership function
(26)$$\mu_{x_{1}}(x_{1})=\left\{ \begin{array}{l} \frac{1}{2}x_{1}-\frac{1}{2}\;\;\;\;1\leq x_{1}\leq 3\\ -\frac{1}{2}x_{1}+\frac{5}{2}\;\;3\leq x_{1}\leq 5 \end{array}\right.$$
and $x_{2}$ and $x_{3}$ are two uncertain-but-bounded variables defined by the following single ellipsoid convex set $x_{2}^{2}+x_{3}^{2}\leq 1$, Table 3, Table 4 and Table 5 give the summarization of the estimation of the membership function of the non-probabilistic reliability index by the single-loop optimization, double-loop optimization and random sampling method, respectively. The comparison of the estimated results is shown in Figure 12.

For this example, it can be seen that the results by single-loop optimization are the best; the ones by the random sampling method and double-loop optimization are almost the same. The computational cost of the single-loop optimization is the least while that of the random sampling method is very large.

### 5.3. A High Nonlinear Performance Function

Suppose a high nonlinear performance function $g(x)=1.016\sqrt{\frac{x_{1}x_{3}^{2}}{x_{2}x_{4}^{4}}}-400$, where $x_{1}$ and $x_{2}$ are two fuzzy variables with the membership functions depicted as follows
(27)$$\mu_{x_{1}}(x_{1})=\left\{ \begin{array}{l} \frac{x_{1}}{0.1\times 10^{7}}-14\;\;1.4\times 10^{7}\leq x_{1}\leq 1.5\times 10^{7}\\ \frac{-x_{1}}{0.2\times 10^{7}}+8.5\;\;1.5\times 10^{7}\leq x_{1}\leq 1.7\times 10^{7} \end{array}\right.$$
(28)$$\mu_{x_{2}}(x_{2})=\left\{ \begin{array}{l} \frac{x_{2}}{0.2\times 10^{-4}}-11.5\;\;2.3\times 10^{-4}\leq x_{2}\leq 2.5\times 10^{-4}\\ \frac{-x_{2}}{0.1\times 10^{-4}}+26\;\;2.5\times 10^{-4}\leq x_{2}\leq 2.6\times 10^{-4} \end{array}\right.$$
and $x_{3}$ and $x_{4}$ are two correlated uncertain-but-bounded variables with the single ellipsoid convex set expressed as
(29)$${\left(\frac{x_{3}-0.98}{4.9\times 10^{-2}}\right)}^{2}+{\left(\frac{x_{4}-20}{1.0}\right)}^{2}\leq 1$$

Table 6, Table 7 and Table 8 give the approximation of the membership function of the non-probabilistic reliability index by the single-loop optimization, double-loop optimization and the random sampling method, respectively. Figure 13 shows a comparison of the estimated results. As can be seen, the random sampling method has achieved the best estimated results but with expensive computational costs, while the single-loop optimization method has given suboptimal results with cheap computation costs. In addition, the single-loop optimization method and random sampling method have achieved good estimated results for the lower bound of the membership function of the non-probabilistic reliability index.

### 5.4. A Cantilever Beam

A cantilever beam subjected to a concentrated force P is shown in Figure 14. The beam has a length of L, a width of b and a height of h. Young’s modulus of the material is E. The structure becomes unsafe when the tip displacement is greater than 0.15 in. Thus, the limit-state function is defined as
(30)$$G=0.15-\frac{4PL^{3}}{Ebh^{3}}$$

In this example, E(psi.), P(lb.) and L(in.) are described by fuzzy variables and the membership functions are expressed as the following three relationships. In addition, b and h are considered to be uncertain-but-bounded variables, and their uncertainty information is summarized in Table 9.
(31)$$\mu_{E}(E)=\left\{ \begin{array}{l} \left(E-0.5\times 10^{7}\right)/5\times 10^{6}\;0.5\times 10^{7}\leq E\leq 1.0\times 10^{7}\\ \left(E-1.1\times 10^{7}\right)/\left(-1\times 10^{6}\right)\;1.0\times 10^{7}\leq E\leq 1.1\times 10^{7} \end{array}\right.;$$
(32)$$\mu_{P}(P)=\left\{ \begin{array}{l} \left(P-50\right)/50\;50\leq P\leq 100\\ \left(P-110\right)/\left(-10\right)\;100\leq P\leq 110 \end{array}\right.;$$
(33)$$\mu_{L}(L)=\left\{ \begin{array}{l} \left(L-29\right)/1\;29\leq L\leq 30\\ \left(L-31\right)/\left(-1\right)\;30\leq L\leq 31 \end{array}\right.;$$

Table 10, Table 11 and Table 12 summarize the results of the membership function by the three methods. A comparison of the estimated results of the non-probabilistic reliability index is shown in Figure 15. As shown in Table 10, Table 11 and Table 12 and Figure 15, the random sampling method has achieved the best estimated result for the membership function of the non-probabilistic reliability index but with heavy computational costs, while the single-loop optimization method has obtained suboptimal results but with low computation costs. In addition, all three methods have achieved good estimated results for the upper bound of the membership function of the non-probabilistic reliability index.

### 5.5. A Performance Function with 12 Variables

The function is given by
(34)$$g(x)=2x_{1}x_{2}x_{3}x_{4}-x_{5}x_{6}x_{7}x_{8}+x_{9}x_{10}x_{11}x_{12}$$
where $x_{i}(i=1,\cdots ,4)$ and $x_{i}(i=5,\cdots ,8)$ are uncertain-but-bounded variables defined by a multi-ellipsoid convex set with two single ellipsoid convex set models, respectively.
(35)$${\sum_{i=1}^{4}\left(\frac{x_{i}-8}{1}\right)}^{2}\leq 1;\;{\sum_{i=5}^{8}\left(\frac{x_{i}-8}{1}\right)}^{2}\leq 1$$
and $x_{i}(i=9,\cdots ,12)$ are fuzzy variables expressed as
(36)$$\mu_{x_{i}}(x_{i})=\left\{ \begin{array}{l} \frac{1}{2}x_{i}-3\;\;6\leq x_{i}\leq 8\\ -\frac{1}{2}x_{i}+5\;\;8\leq x_{i}\leq 10 \end{array}\right.,(i=9,\cdots ,12)$$

Table 13, Table 14 and Table 15 give the corresponding estimated results of the membership functions. The estimated membership functions of the non-probabilistic reliability index are given in Figure 16. The results show that the three methods have achieved good estimated results, and the computational consumption cost of the single-loop optimization is the least. The random sampling method is not practical for many engineering applications owing to the prohibitive computational cost. The results show that three methods can achieve good results for the nonlinear performance function with multiplication operation, addition operation and subtraction operation.

## 6. Conclusions

This paper investigates the uncertainty propagation for systems with fuzzy variables and uncertain-but-bounded variables, and the membership function of the non-probabilistic reliability index is employed to define the uncertainty propagation. The proposed methods can be applied for uncertain analysis of any systems (such as structures and machines) that are subjected to uncertain factors with limited samples and uncertainty relevant to human knowledge and expert experience. Three algorithms, namely, single-loop optimization, double-loop optimization and random sampling are proposed to solve the membership function of the non-probabilistic reliability index. Five examples with linear and nonlinear problems are employed to demonstrate the applicability of the proposed methods. The results show that single-loop optimization is more efficient and stable than double-loop optimization. Although the results using the random sampling method are better than those using single-loop optimization for most cases, the former approach is not suitable for many engineering applications due to its huge computation cost. In addition, the results also show that the single loop method fits to the linear performance functions and nonlinear performance functions with multiplication operation, addition operation and subtraction operation. Meanwhile, the computational cost relevant to the single-loop optimization method can be afforded as the number of uncertainties increases. The main contribution of the proposed method is to propose a model to deal with the uncertainty analysis for systems with fuzzy variables and uncertain-but-bounded variables and to give three methods for solving this issue. Future research can focus on the more adaptable approaches that can find the global optimal solutions for linear performance functions and nonlinear performance functions.

## Figures and Tables

**Figure 1 materials-16-03367-f001:**
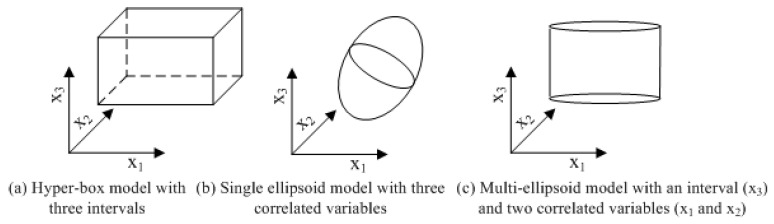
The comparison of hyper-box model, single ellipsoid model and multi-ellipsoid model.

**Figure 2 materials-16-03367-f002:**
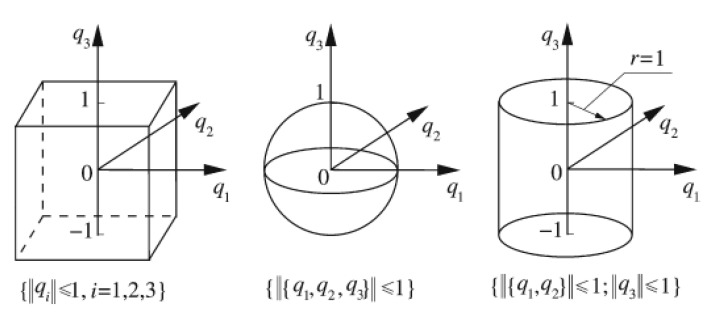
Comparison of hyper-box model, single ellipsoid model and multi-ellipsoid model in normalized space.

**Figure 3 materials-16-03367-f003:**
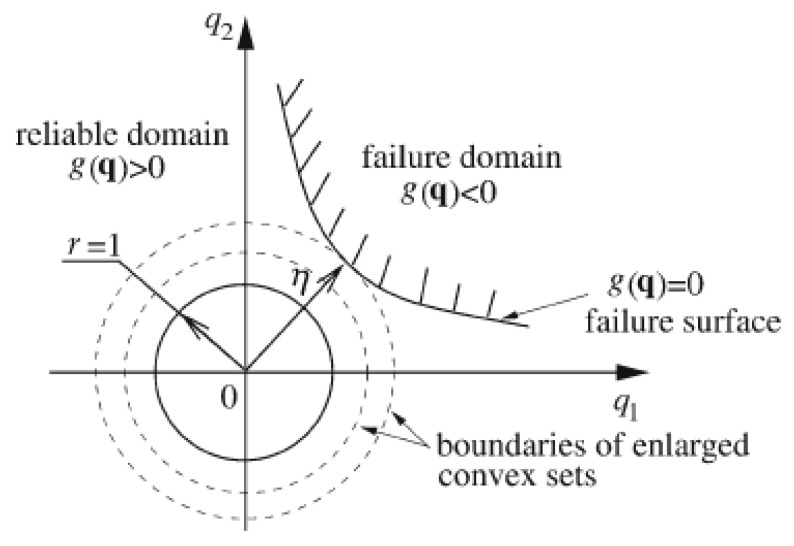
Non-probabilistic reliability index in case of a single two-dimensional ellipsoid convex set model.

**Figure 4 materials-16-03367-f004:**
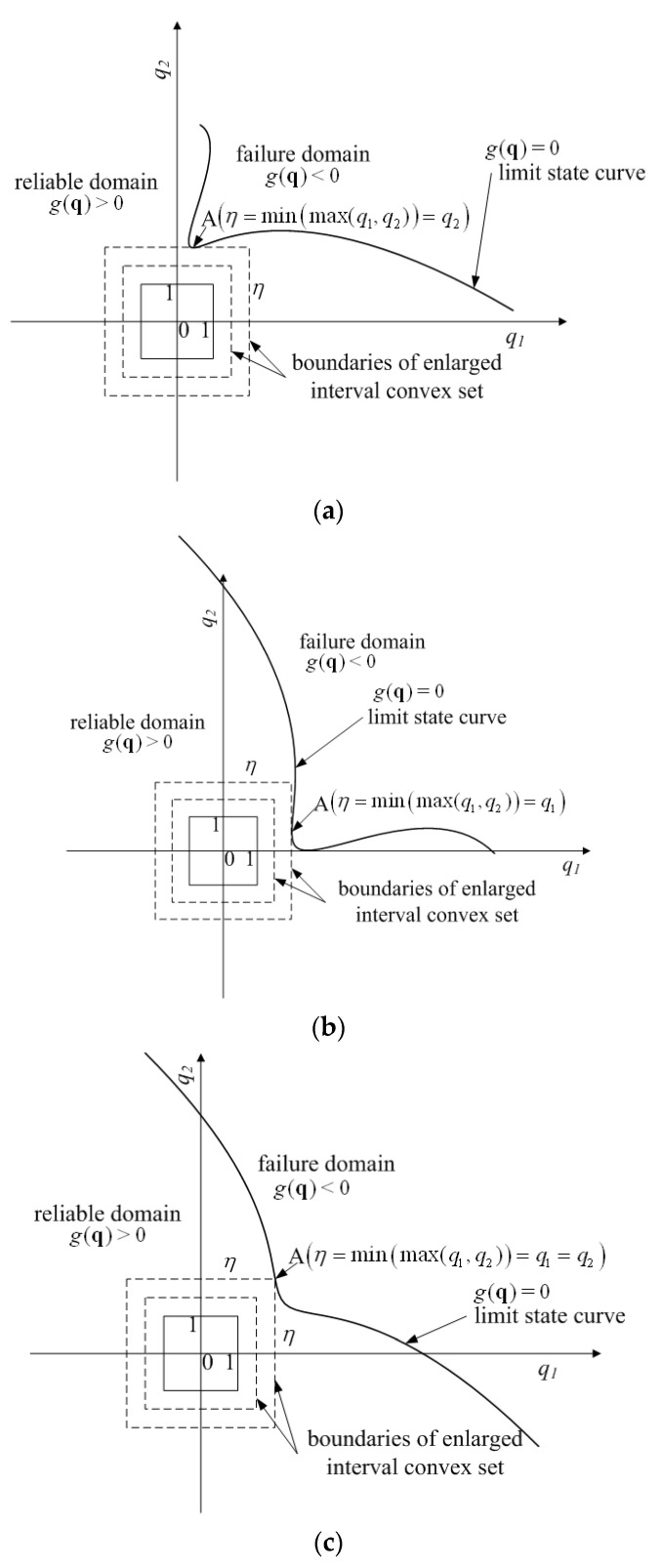
Three cases for non-probabilistic reliability index in case of two interval variables. (**a**) The normalized limit state curve intersects with the enlarged box at one side. (**b**) The normalized limit state curve intersects with the enlarged box at another side. (**c**) The normalized limit state curve intersects with the enlarged box at the cater-corner point.

**Figure 5 materials-16-03367-f005:**
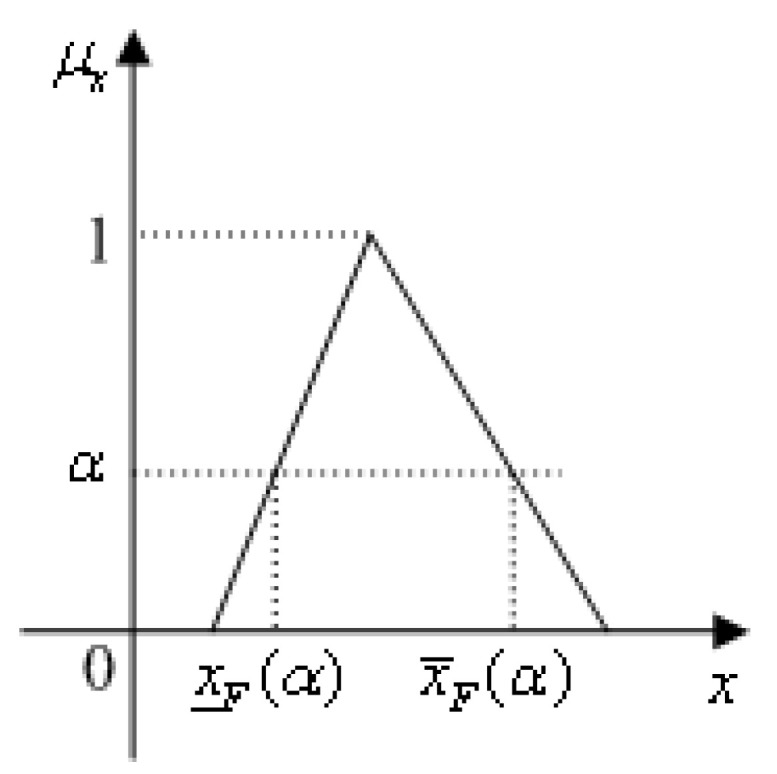
A fuzzy variable reduces to an interval at the membership level α.

**Figure 6 materials-16-03367-f006:**
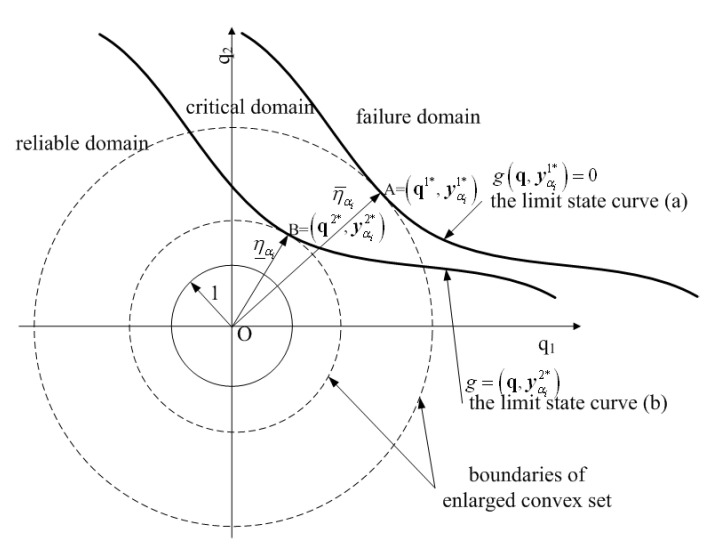
Schematic representation of the non-probabilistic reliability index for the model with a two-dimensional single ellipsoid convex set and fuzzy variables in the q-space.

**Figure 7 materials-16-03367-f007:**
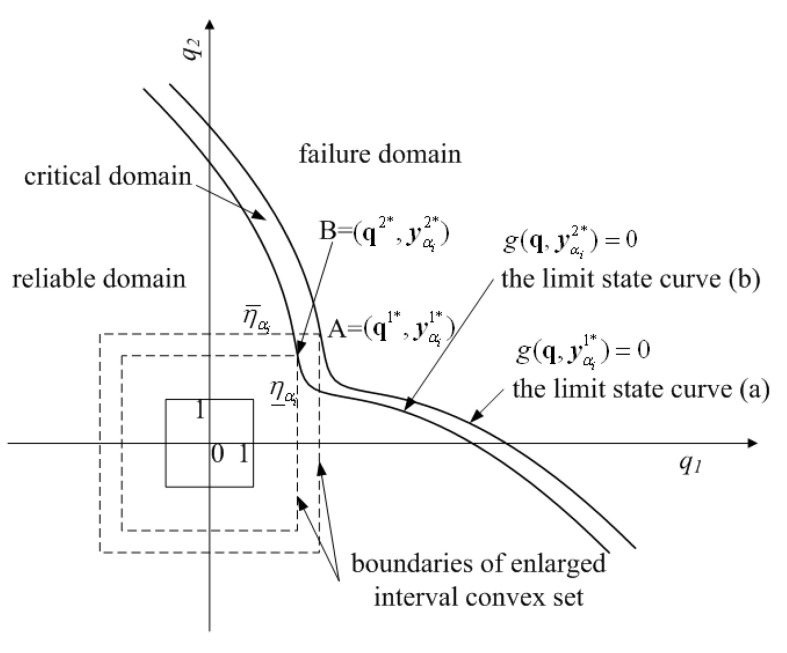
Schematic representation of the non-probabilistic reliability index for the model with two interval variables and fuzzy variables in the q-space.

**Figure 8 materials-16-03367-f008:**
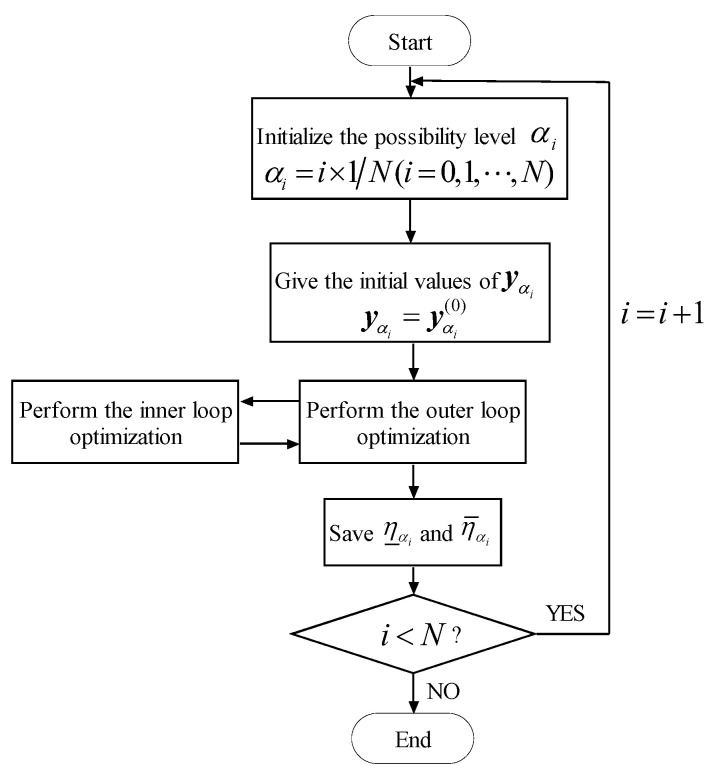
Estimate the ${\underline{\eta }}_{\alpha_{i}}$ and ${\bar{\eta }}_{\alpha_{i}}$ by using double-loop optimization.

**Figure 9 materials-16-03367-f009:**
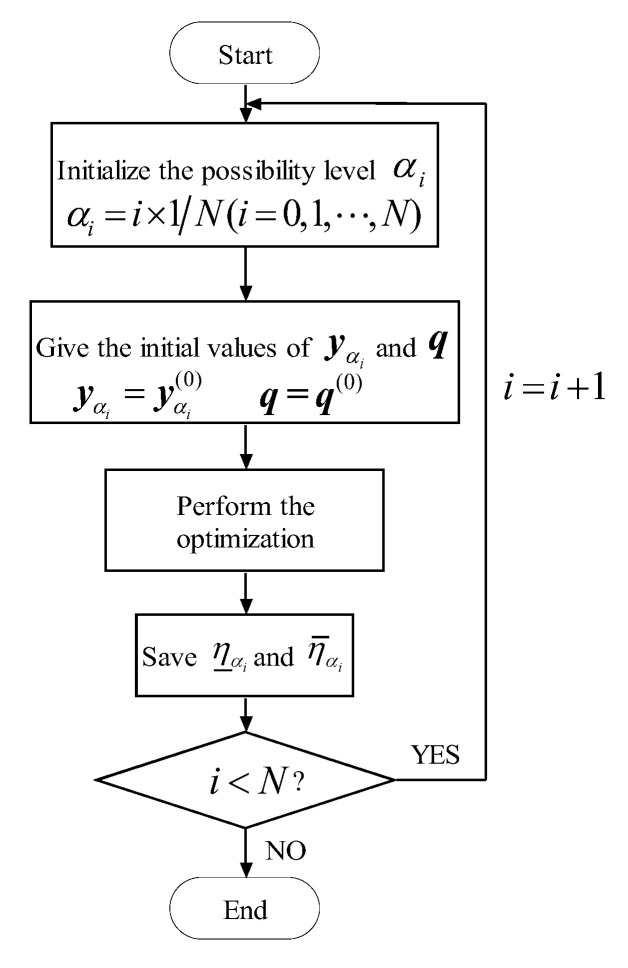
Estimate ${\underline{\eta }}_{\alpha_{i}}$ and ${\bar{\eta }}_{\alpha_{i}}$ by using single-loop optimization.

**Figure 10 materials-16-03367-f010:**
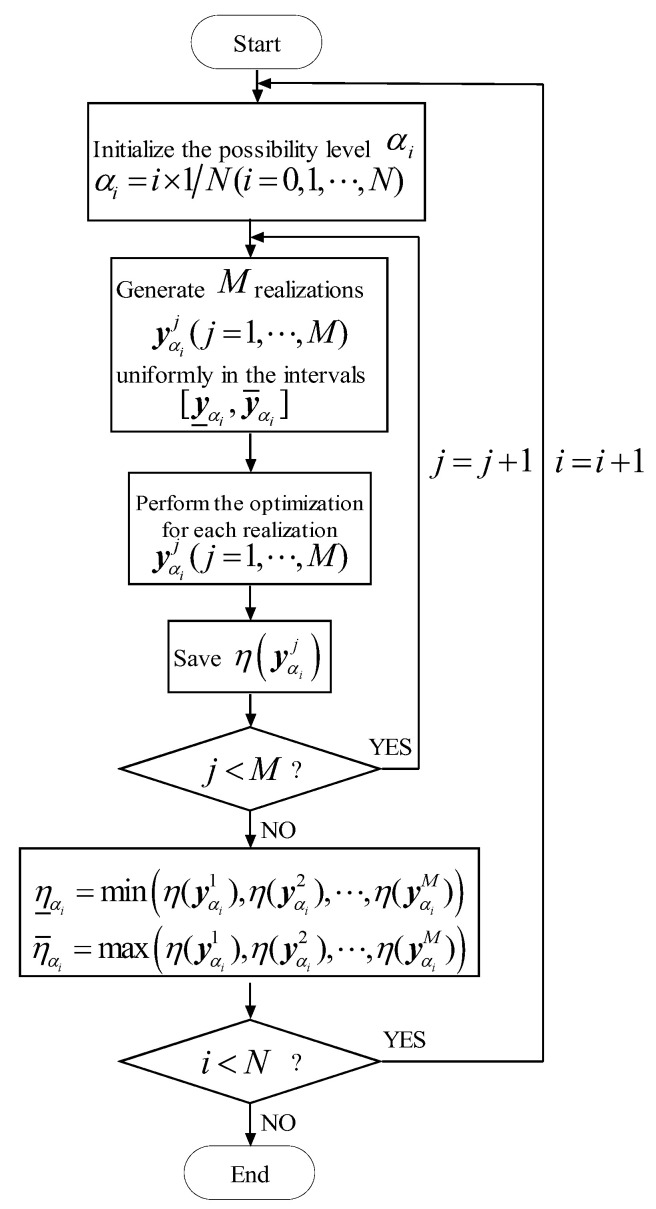
Estimate the ${\underline{\eta }}_{\alpha_{i}}$ and ${\bar{\eta }}_{\alpha_{i}}$ by the random sampling method.

**Figure 11 materials-16-03367-f011:**
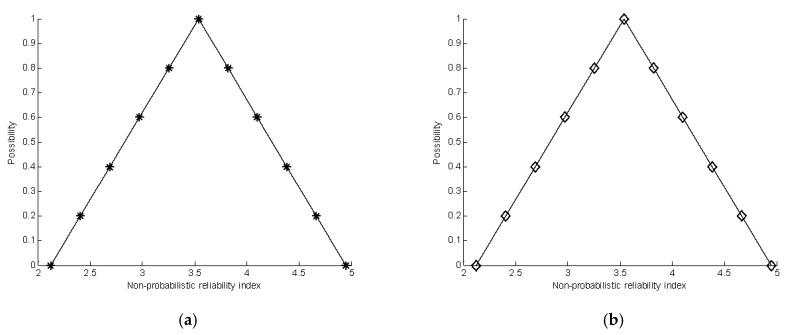
Comparison of the calculation of the membership function of the non-probabilistic reliability index. (**a**) The membership function of the non-probabilistic reliability index by single-loop optimization. (**b**) The membership function of the non-probabilistic reliability index by double-loop optimization.

**Figure 12 materials-16-03367-f012:**
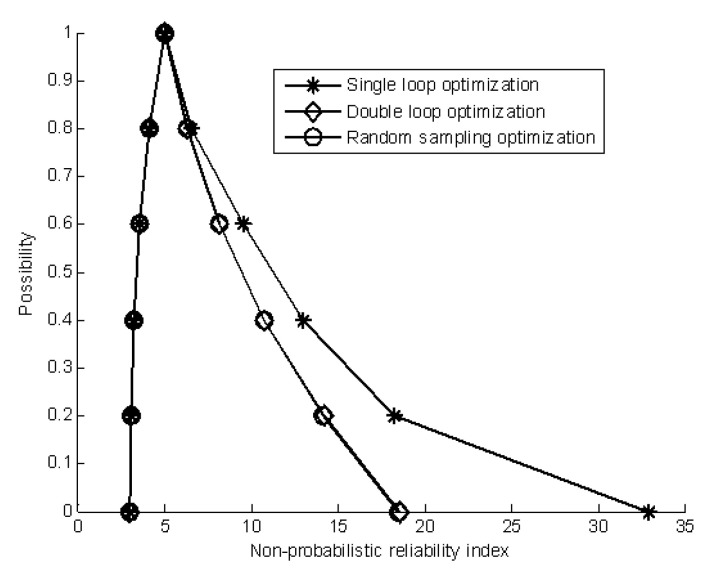
The estimated membership function of the non-probabilistic reliability index.

**Figure 13 materials-16-03367-f013:**
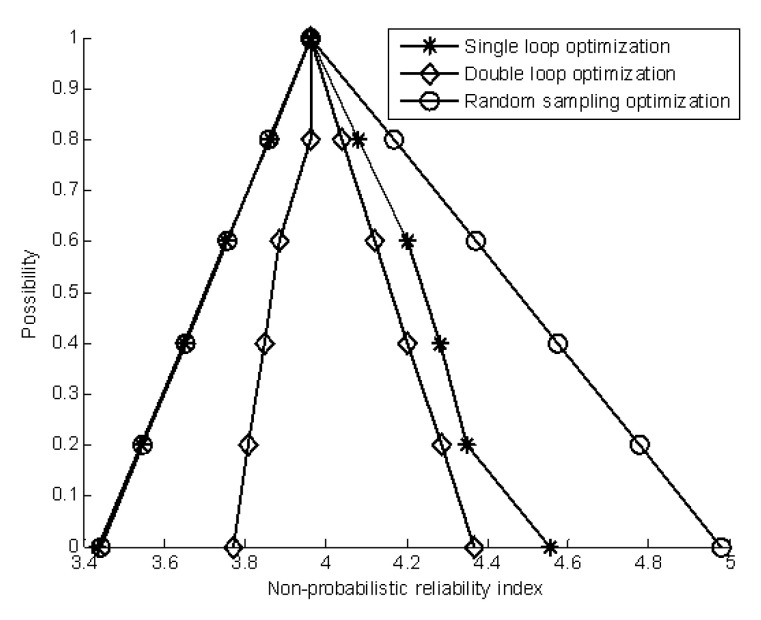
The estimated membership function of the non-probabilistic reliability index.

**Figure 14 materials-16-03367-f014:**
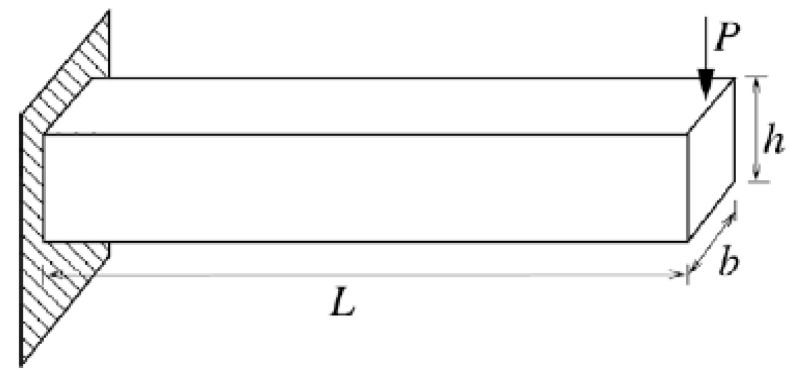
A cantilever beam.

**Figure 15 materials-16-03367-f015:**
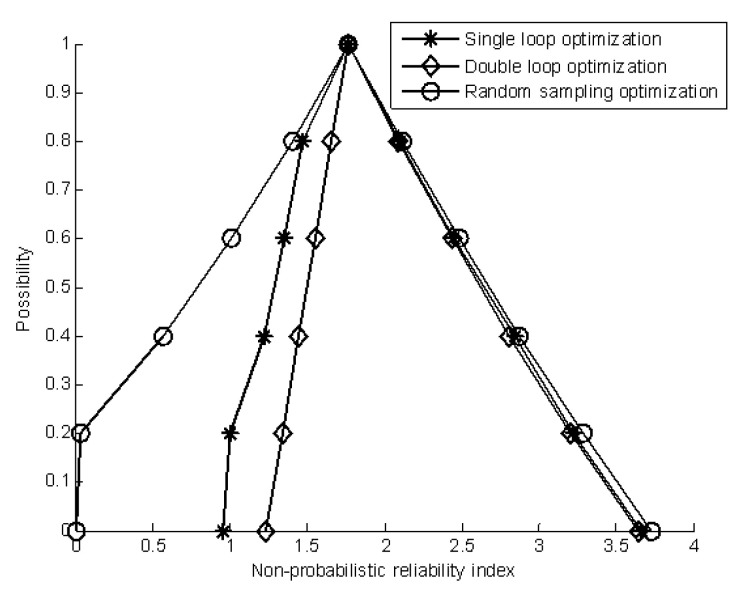
The estimated membership function of the non-probabilistic reliability index.

**Figure 16 materials-16-03367-f016:**
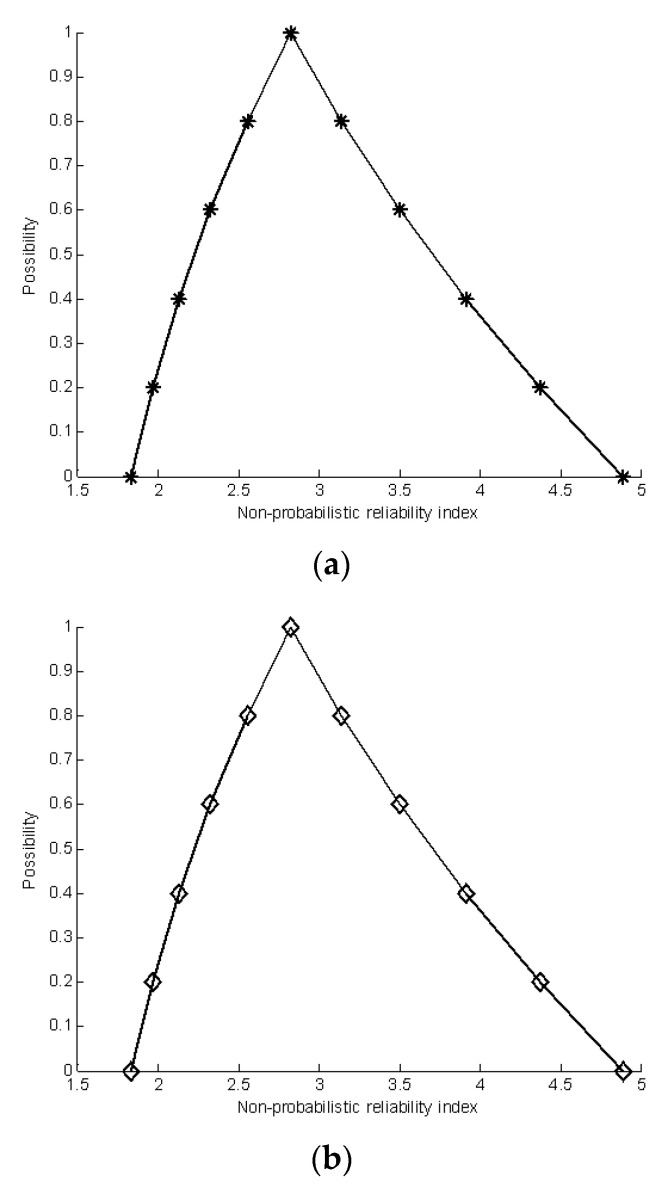
Comparison of estimated results by three methods; (**a**) The estimated membership function by using single-loop optimization; (**b**) The estimated membership function by using double-loop optimization; (**c**) The estimated membership function by using random sampling optimization.

**Table 1 materials-16-03367-t001:** The estimated results by using single-loop optimization.

	0.0	0.2	0.4	0.6	0.8	1.0
NOFC	8 + 20	8 + 16	8 + 12	8 + 12	8 + 12	8
$\underline{\eta }$	2.1213	2.4042	2.6870	2.9698	3.2527	3.5355
$\bar{\eta }$	4.9497	4.6669	4.3841	4.1012	3.8184	3.5355

**Table 2 materials-16-03367-t002:** The estimated results by using double-loop optimization.

	0.0	0.2	0.4	0.6	0.8	1.0
NOFC	36 + 36	24 + 36	24 + 36	24 + 36	24 + 36	24
$\underline{\eta }$	2.1213	2.4042	2.6870	2.9698	3.2527	3.5355
$\bar{\eta }$	4.9497	4.6669	4.3841	4.1012	3.8184	3.5355

**Table 3 materials-16-03367-t003:** The estimated results by using the single-loop optimization.

	0.0	0.2	0.4	0.6	0.8	1.0
NOFC	53 + 405	57 + 545	112 + 518	151 + 362	115 + 398	135
$\underline{\eta }$	3.0250	3.0960	3.2624	3.5856	4.1424	5.0250
$\bar{\eta }$	32.9722	18.2668	13.0190	9.5960	6.5653	5.0250

**Table 4 materials-16-03367-t004:** The estimated results by using the double-loop optimization.

	0.0	0.2	0.4	0.6	0.8	1.0
NOFC	216 + 611	156 + 372	162 + 734	196 + 386	238 + 422	135
$\underline{\eta }$	3.0250	3.0960	3.2624	3.5856	4.1424	5.0250
$\bar{\eta }$	18.6250	14.1936	10.7792	8.2128	6.3408	5.0250

**Table 5 materials-16-03367-t005:** The estimated results by random sampling.

	0.0	0.2	0.4	0.6	0.8	1.0
NOFC	66,472	65,471	65,481	65,066	65,616	71
$\underline{\eta }$	3.0254	3.0969	3.2638	3.5873	4.1438	5.0250
$\bar{\eta }$	18.5276	14.1329	10.7446	8.1957	6.3347	5.0250

**Table 6 materials-16-03367-t006:** Summary of estimated results using single-loop optimization.

	0	0.2	0.4	0.6	0.8	1.0
NOFC	62 + 85	70 + 91	69 + 84	83 + 95	454 + 121	90
$\underline{\eta }$	3.4356	3.5418	3.6475	3.7527	3.8609	3.9618
$\bar{\eta }$	4.5565	4.3511	4.2831	4.2012	4.0819	3.9618

**Table 7 materials-16-03367-t007:** Summary of estimated results using double-loop optimization.

	0	0.2	0.4	0.6	0.8	1.0
NOFC	150 + 150	150 + 150	150 + 150	150 + 150	75 + 150	75
$\underline{\eta }$	3.7706	3.8082	3.8461	3.8844	3.9618	3.9618
$\bar{\eta }$	4.3695	4.2851	4.2021	4.1206	4.0405	3.9618

**Table 8 materials-16-03367-t008:** Summary of estimated results using random sampling.

	0	0.2	0.4	0.6	0.8	1.0
NOFC	75,892	77,724	79,897	81,096	82,208	83
$\underline{\eta }$	3.4419	3.5468	3.6512	3.7552	3.8587	3.9618
$\bar{\eta }$	4.9787	4.7778	4.5758	4.3725	4.1679	3.9618

**Table 9 materials-16-03367-t009:** Information of uncertain-but-bounded variables for a cantilever beam.

Uncertain Variable	Nominal Value	Convex Model Description
b(in.)	0.8359	$\left[\begin{array}{cc} \delta_{b} & \delta_{h} \end{array}\right]\left[\begin{array}{cc} 1 & 0\\ 0 & 1 \end{array}\right]\left[\begin{array}{c} \delta_{b}\\ \delta_{h} \end{array}\right]\leq 0.1^{2}$
h(in.)	2.5093	

**Table 10 materials-16-03367-t010:** Summary of results using single-loop optimization.

	0	0.2	0.4	0.6	0.8	1.0
NOFC	185 + 224	184 + 220	175 + 215	174 + 205	182 + 204	35
$\underline{\eta }$	0.9558	1.001	1.2234	1.3521	1.4675	1.7622
$\bar{\eta }$	3.6652	3.2254	2.8415	2.4511	2.1012	1.7622

**Table 11 materials-16-03367-t011:** Summary of results using double-loop optimization.

	0	0.2	0.4	0.6	0.8	1.0
NOFC	484 + 7067	484 + 2944	484 + 2302	502 + 1693	372 + 968	22
$\underline{\eta }$	1.2338	1.3394	1.4451	1.5510	1.6570	1.7632
$\bar{\eta }$	3.6444	3.2065	2.8069	2.4370	2.0906	1.7632

**Table 12 materials-16-03367-t012:** Summary of results using random sampling.

	0	0.2	0.4	0.6	0.8	1.0
NOFC	61,820	57,345	57,067	56,263	54,650	60
$\underline{\eta }$	0.00777	0.03082	0.56639	1.01325	1.4083	1.7624
$\bar{\eta }$	3.72747	3.28465	2.87335	2.48583	2.1168	1.7624

**Table 13 materials-16-03367-t013:** Summary of results using single-loop optimization.

	0	0.2	0.4	0.6	0.8	1.0
NOFC	164 + 254	134 + 239	149 + 254	224 + 224	149 + 224	149
$\underline{\eta }$	1.8367	1.9713	2.1341	2.3289	2.5594	2.8297
$\bar{\eta }$	4.8907	4.3767	3.9147	3.5041	3.1435	2.8297

**Table 14 materials-16-03367-t014:** Summary of results using double-loop optimization.

	0	0.2	0.4	0.6	0.8	1.0
NOFC	1745 + 1855	1690 + 1910	1745 + 1800	1690 + 1635	1745 + 1745	109
$\underline{\eta }$	1.8367	1.9713	2.1341	2.3289	2.5594	2.8297
$\bar{\eta }$	4.8907	4.3767	3.9147	3.5042	3.1435	2.8297

**Table 15 materials-16-03367-t015:** Summary of results using random sampling.

	0	0.2	0.4	0.6	0.8	1.0
NOFC	106,468	105,187	101,965	101,722	100,016	131
$\underline{\eta }$	1.8378	1.9724	2.1352	2.3297	2.5599	2.8297
$\bar{\eta }$	4.8802	4.3691	3.9096	3.5012	3.1421	2.8297

## Data Availability

Not applicable.

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
