# Peer review of "Uncertainty Propagation for the Structures with Fuzzy Variables and Uncertain-but-Bounded Variables"

_materials, 2023, doi:10.3390/ma16093367_

Round 1

Reviewer 1 Report

The present paper deals with problems in presence of uncertain variables modelled by fuzzy variables and uncertain-but-bounded variables.

Specifically the authors investigate the uncertainty propagation problem with fuzzy variables described by membership function and uncertain-but-bounded variables defined  by multi-ellipsoid convex set.

First the authors frame the problem with a detailed description of non-probabilistic convex set models with the proper introduction of the ellipsoid convex set model to take into account the correlation among the uncertain-but-bounded parameters.

By defining the non-probabilistic reliability index for uncertain-but-bounded variables and combining with the membership levels method for fuzzy variables, three methods are proposed to solve the uncertainty propagation namely seeking the membership function of the non-probabilistic reliability index.

The paper is of practical interest due to the circumstance outlined by the authors that many engineering problems involve uncertain parameters arising from different sources and with lack of information such as to prevent a probabilistic description in terms of probability density function.

The accuracy of the derived methods is verified in representative examples.

In the reviewer's opinion, two points should be deepened and extended.

The first point concerns the computational cost as the number of uncertainties increases. In the numerical examples, several cases are explored considering (except for the last case) a small number of uncertain variables.

The second point concerns the fact that the title as well as the main test of the paper refer to structures and practical engineering problems.

However, with the exception of the cantilever beam problem, any other structure has been considered in the proposed analysis. An engineering application should be added or it is incorrect to refer to “structures”.

As minor comment, please check the position of Eq.11 in the text.

Therefore, I agree that the paper is suitable for publication in Material, on the understanding that the authors will elaborate the manuscript complying with the points raised above.

Reviewer 2 Report

In this paper, the authors focus on the on the practical structures subjected to epistemic uncertainty measured using fuzzy variable and uncertainty with limited samples measured using uncertain-but-bounded variables. Multi-ellipsoid convex set is used to investigate the uncertainty propagation of the structures with fuzzy variables described by membership function and uncertain-but-bounded variables. The present methodology suggested in this work can applied to investigate in wide range of problems in the structure testing, design and fabrication.

The accuracy of the present methodology is confirmed via some standard test problems with different order, linear and nonlinear of polynomial performance function. Then the proposed methodology and program is applied to investigate some real problems. The manuscript contains some merits, so it can be reconsidered after carefully resolving the following issues:

(1) The application of fuzzy in the structures analysis have investigated by many authors, some following works on the application of fuzzy description and membership function can be considered and discussed in introduction:

https://doi.org/10.1016/j.ijar.2018.04.013

https://doi.org/10.1016/j.finel.2011.09.012

https://doi.org/10.1016/j.tws.2014.09.003

(2) The results presented in section 5.4 for a cantilever beam should more discussed in details according to Tables 10-12 and Figure 15.

(3) The result of the example given in section 5.5 for 12 variables function also need more explained in details to give more valuable conclusion of the present methodology. Also, the potential application of the present methodology should be given in the conclusion section.

(4) The conclusion section should be re-written, the reason of the research is current given in abstract section, so it is not necessary to repeat again.

Round 2

Reviewer 2 Report

The manuscript is revised well, all suggestions have been addressed. Hence it can be accepted for publication.
